# Comprehensive analysis of differentially expressed miRNAs in hepatocellular carcinoma: Prognostic, predictive significance and pathway insights

**Kayleigh Smith**[1‡], **Dan Beach**[2‡], **Roger Silva**[3], **Gyorffy Balazs**[4,7], **Francesca Salani**[1,5,6‡]*, **Francesco Crea**[1‡]

**1** Cancer Research Group-School of Life Health and Chemical Sciences, The Open University, Milton Keynes, United Kingdom, **2** School of Life Sciences, University of Sussex, Brighton, United Kingdom, **3** Department of Medicine, Cancer Research Program Research Institute of the McGill University Health Centre, McGill University, Montreal, Canada, **4** Department of Bioinformatics, Semmelweis University, Budapest, Hungary, **5** Department of Translational Research and New Technologies in Medicine and Surgery, University of Pisa, Pisa, Italy, **6** Institute of Interdisciplinary Research "Health Science", Scuola Superiore Sant'Anna, Pisa, Italy, **7** Research Centre for Natural Sciences, Institute of Molecular Life Sciences, Budapest, Hungary

‡ KS and DB are contributed equally to this work as co first authors. FS and FC are contributed equally to this work as co last authors.
* f.salani1@gmail.com

**Data Availability Statement:** Data are publicly available from the repositories referenced in the manuscript.

## Abstract

Robust prognostic and predictive factors for hepatocellular carcinoma, a leading cause of cancer-related deaths worldwide, have not yet been identified. Previous studies have identified potential HCC determinants such as genetic mutations, epigenetic alterations, and pathway dysregulation. However, the clinical significance of these molecular alterations remains elusive. MicroRNAs are major regulators of protein expression. MiRNA functions are frequently altered in cancer. In this study, we aimed to explore the prognostic value of differentially expressed miRNAs in HCC, to elucidate their associated pathways and their impact on treatment response. To this aim, bioinformatics techniques and clinical dataset analyses were employed to identify differentially expressed miRNAs in HCC compared to normal hepatic tissue. We validated known associations and identified a novel miRNA signature with potential prognostic significance. Our comprehensive analysis identified new miRNA-targeted pathways and showed that some of these protein coding genes predict HCC patients' response to the tyrosine kinase inhibitor sorafenib.

## 1. Introduction

Primary liver cancer is the second most common cause of cancer mortality in the world [1]. Hepatocellular carcinoma (HCC) is the most common histology identified in primary liver cancers. HCC prognosis is primarily determined by stage, liver function, patient conditions and, indirectly, by HCC aetiology [2]. The five-year survival ranges from more than 50% in

**Funding:** The author(s) received no specific funding for this work.

**Competing interests:** The authors have declared that no competing interests exist.

operable diseases to less than 5% in the most advanced stages [3]. This is due to the lack of effective treatments, especially in the setting of high tumour burden, where systemic treatments can lead to a median survival of less than two years. Despite extensive molecular studies which have identified potential determinants among genetic mutations, epigenetic alterations and pathway dysregulation [4], HCC pathogenesis is not entirely elucidated. A better understanding of the molecular processes driving HCC could lead to more effective treatments.

MicroRNAs (miRNAs) are short non-coding RNAs that regulate mRNA translation and stability [5]. The human genome encodes approximately 2000 miRNAs, but each of these transcripts can regulate the expression of hundreds of mRNAs. It has been estimated that miRNAs regulate the expression of approximately 60% of the protein-coding genes. The expression of miRNAs is deregulated in many cancers [6], making them attractive biomarkers and therapeutic targets. The role of specific miRNAs in HCC has been studied [7], but there are only limited experiences on comprehensive studies assessing the prognostic function of miRNAs in HCC and the downstream pathways in this malignancy [8–10] leading to the identification of non-univocal signatures.

In this study, we used a combination of bioinformatic techniques and clinical dataset analyses to identify miRNAs that are differentially expressed in HCC *versus* normal hepatic tissue and whose expression is associated with HCC prognosis. This approach allowed us to confirm known associations, identify several uncharacterised miRNAs and propose new potential targeting pathways with potential clinical relevance.

## 2. Material and methods

### 2.1 Differential expression analysis

A data-led approach was chosen over a literature-based candidate gene-identifying approach for the sake of analysis comprehensiveness. OncoMir Cancer Database (OMCD) [11] was used to access data on miRNA expression from Liver hepatocellular carcinoma (referred to as LIHC) dataset. LIHC dataset comprises The Cancer Genome Atlas (TCGA) data from 426 samples, of which 375 HCC patients and 51 non-HCC liver tissue. Differential expression between HCC and non-HCC liver tissue was tested via *t-test* on OMCD portal (Bonferroni-adjusted p values of $< 0.05$) and miRNAs whose expression in HCC was at least 2 folds different (ratios $> 2$ or $< 0.5$) from non-HCC were retained for further analyses.

### 2.2 Survival analysis

The shortlisted differentially expressed miRNAs were cross-referenced with miRNAs RNA-seq TCGA dataset provided by Kaplan-Meier Plotter (KMP) tool [12]. Only RNA-seq data from TCGA samples of patients who had not undergone any other treatment (either radio- or chemotherapy or systemic drugs) than surgery was comprised in the survival analysis. Patients were dichotomized according to each differentially expressed miRNA median values, into high- and low-expression cohorts. Overall survival (OS) was chosen as the prognostic endpoint of interest and compared among high- and low-expression patients through log-rank test p-value. Each comparison is presented as median OS (mOS), Hazard Ratio (HR) with 95% Interval of Confidence (95%IC) and p-value.

### 2.3 Construction of a prognostic miRNA signature

MiRNAs identified in 2.2 bearing a significant prognostic value were used to generate a prognostic signature through miRpower tool in the KMP. "*Multiple genes*" and "*mean gene*

*expression*" options were set; patients were dichotomized according to the median value of the signature and the OS of the two cohorts was compared via log-rank test.

## 2.4 Pathway analysis

Pathway analysis was conducted to explore the mechanisms underpinning the prognostic effects of these differentially expressed miRNAs. This involved the identification of target genes for up- and down-regulated miRNAs, and KEGG pathway analysis using the Over Representation Analysis (ORA) method.

**2.4.1 Selection of mature miRNAs.** The TCGA miRNA expression data used by the OMCD and KM Plotter databases do not differentiate between mature 3p and 5p miRNAs [13]. Both miRNA forms are capable of binding to target mRNAs through a sequence-specific base pairing mechanism and regulate gene expression by interacting with distinct sets of target genes. However, only one mature strand is typically incorporated into the RNA-induced silencing complex (RISC) and involved in regulating mRNA transcription. Thus, for each miRNA, it was necessary to determine whether the 3p or 5p form should be used for the analysis of downstream target genes and pathways. The predominant mature form of each miRNA was selected based on the number of experimental reads in the miRBase database [14] (**S1 Table**). The *hsa-miR*-3653 identifier was subsequently excluded as the available experimental data no longer supports its annotation as a miRNA [14].

**2.4.2 Target gene identification.** The first step in conducting the pathway analysis was to choose a database that provides miRNA-target interaction data. MirTarBase is a manually curated database of experimentally validated miRNA-target interactions (MTI) collated from reporter assay, western blot and next generation sequencing (NGS) studies [15]. The database includes more than 360,000 MTIs and represents the largest and most up to date source of experimentally validated MTI data. MirTarBase was initially selected due to its size and rigorous inclusion criteria. An alternative MTI database, miRPathDB was found to include the latest version mirTarBase data and was selected instead. The prognostic mature miRNA identifiers were queried in the mirRPathDB database, retrieving a comprehensive list of target genes. This list was cross-validated using the miRTargetLink 2.0 database which provided a second mirror of the most recent mirTarBase data, confirming a 100% match between the predictions provided by the two databases.

A Python script was written to remove duplicates and genes with no experimental evidence of miRNA-interaction. Two genes could not be mapped to current Ensembl/Entrez IDs and were excluded from further analysis. The remaining target genes were relevant and aligned with the objective of the analysis.

**2.4.3 Functional annotation and enrichment analysis.** Functional annotation and enrichment analysis was conducted using the Database for Annotation, Visualization and Integrated Discovery (DAVID) tool. Alternative online functional enrichment tools WebGestalt and g:Profiler were evaluated, but DAVID was selected because it is highly cited, regularly updated, and well documented [16]. The target gene lists for up-regulated miRNAs and down-regulated miRNAs were uploaded to DAVID which identified them as being for the *Homo Sapiens* species. Functional annotation enrichment was performed for targets of upregulated miRNAs and downregulated miRNAs separately, selecting only the KEGG pathway database option. Statistical analysis tested the hypothesis that the overlap between genes in the target gene list and a given pathway or functional category was greater than expected by chance. The statistical test used was Fisher's exact test and Benjamini-Hochberg correction was applied to p-values to correct for multiple tests. Text-based result files for each gene list were downloaded

from DAVID. A Python script was written to filter pathways with a Benjamini-Hochberg significance threshold of $\leq 0.05$.

**2.4.4 Analysis of enriched KEGG pathways.** To focus on the most relevant pathways, KEGG analysis results were filtered to include only pathways that are putatively involved in HCC. Relevant pathways were defined as those listed as "related" in the KEGG HCC Pathway (*hsa05225*) [17]. A Python script was used to filter the KEGG results according to a predefined set of keywords derived from these maps. The keyword list used to filter results was: Hepatocellular, Hepatitis, Alcohol, NAFLD, *PI3K*, *AKT*, *P53*, *WNT*, *MAPK*, Cell cycle, calcium signaling, *TGF-beta*.

**2.4.5 Validation of the functional enrichment analysis.** A validation of the DAVID functional enrichment analysis was performed using the WebGeSTALT tool, accessed in February 2024 [18].

**2.4.6 Identifying potential miRNA interactions with cancer driver genes.** A previously reported pan-cancer analysis of more than 9000 tumor exomes from 33 TCGA projects identified 299 oncogenic driver genes and 3473 associated driver missense mutations [19]. Subsequent functional validation utilizing an independent dataset of experimentally validated mutations indicated that 60–85% of the predicted mutations were likely to be cancer drivers. A list of 299 HCC and pan-cancer associated genes was retrieved from TCGA and a Python script was used to return intersection between the prognostic miRNA target genes and the 299 putative cancer driver genes.

## 3. Results

### 3.1 Some differentially expressed miRNAs are prognostic in resected HCC

Using OMCD, levels of expression of miRNAs in HCC *versus* healthy liver tissue were compared. Out of the 106 significantly differentially expressed miRNAs, 59 had a significant, higher than 2-fold differential expression (HCC/healthy liver either >2 or <0.5), compared to normal liver: respectively, 23 were increased and 36 decreased in HCC (**S2 Table**). Forty of the differentially expressed miRNAs were also present in the KMP RNA-seq dataset which we have employed for prognostic stratification (21 of which up-regulated and 19 down-regulated in HCC *versus* healthy liver).

Of the 40 shortlisted miRNAs, 10 showed a statistically significant correlation with OS (p for log-rank test <0.05); in particular, 5 among the HCC up-regulated (*hsa-miR*-501, *hsa-miR*-877, *hsa-miR*-1180, *hsa-miR*-3127, *hsa-miR*-3677) and 5 among the HCC down-regulated miRNAs (*hsa-miR*-3653, *hsa-miR*-99a, *hsa-miR*-145, *hsa-miR*-326, *hsa-miR*-139) (**Table 1**).

In keeping with our hypothesis, we found that for the majority of miRNAs, expression in HCC *versus* normal tissue and prognostic significance were highly correlated. All the miRNAs up-regulated in HCC were associated with negative prognosis, 4 out of 5 miRNAs that are down-regulated in HCC were associated with better prognosis. *Hsa-miR*-326 was the only transcript decreased in HCC showing an unexpected negative prognostic role (HR:1.97(1.37–2.83), p 0.0002).

To increase the robustness of our analysis, a 9-miRNA prognostic signature was generated from prognostically relevant miRNAs (this signature excluded *Hsa-miR*-326, which showed inconsistent correlations between prognosis and expression patterns in normal vs neoplastic liver tissues). The 9-miRNA signature yielded an HR for OS of 2.15 (1.52–3.05), with log-rank p of 0.00005 (**Fig 1A**), consistent across different clinical sub-groups, as shown in **Fig 1B–1F**.

The prognostic significance in terms of OS of each miRNA belonging to the signature is plotted separately in S2 Fig.

**Table 1. OS correlation of differentially expressed miRNAs (> 2-fold increase or decrease in HCC vs normal livers), according to KMP; miRNAs in bold are significantly correlated with OS.**

| MiRNA identifier | P value | HR | Median survival—Low expression (Months) | Median survival -High expression (Months) |
|---|---|---|---|---|
| | | Increased expression in HCC | | |
| hsa-miR-660 | 0.6338 | 1.09 (0.77–1.54) | 58.88 | 55.4 |
| hsa-miR-581 | 0.4568 | 1.14 (0.81–1.61) | 55.69 | 53.33 |
| hsa-miR-93 | 0.455 | 1.14 (0.81–1.62) | 55.69 | 80.75 |
| hsa-miR-10b | 0.2934 | 1.21 (0.85–1.71) | 55.4 | 58.88 |
| hsa-miR-532 | 0.2351 | 1.24 (0.87–1.75) | 55.4 | 83.24 |
| hsa-miR-34a | 0.2247 | 0.81 (0.57–1.14) | 55.4 | 58.88 |
| hsa-miR-183 | 0.1846 | 1.27 (0.89–1.79) | 70.06 | 45.93 |
| hsa-miR-500b | 0.1732 | 1.27 (0.9–1.81) | 70.06 | 45.57 |
| hsa-miR-96 | 0.1296 | 1.31 (0.92–1.85) | 70.06 | 45.93 |
| hsa-miR-224 | 0.1179 | 1.32 (0.93–1.88) | 70.06 | 46.78 |
| hsa-miR-222 | 0.0882 | 1.35 (0.95–1.92) | 70.06 | 51.29 |
| hsa-miR-589 | 0.0865 | 1.36 (0.96–1.92) | 60.89 | 51.29 |
| hsa-miR-221 | 0.0696 | 1.38 (0.97–1.96) | 70.06 | 51.29 |
| hsa-miR-500a | 0.0666 | 1.38 (0.98–1.96) | 69.57 | 58.88 |
| hsa-miR-452 | 0.0657 | 1.39 (0.98–1.97) | 80.75 | 45.93 |
| hsa-miR-21 | 0.0552 | 1.41 (0.99–2) | 70.06 | 46.78 |
| **hsa-miR-501** | **0.05** | **1.42 (1–2.01)** | **69.57** | **55.4** |
| **hsa-miR-877** | **0.0158** | **1.54 (1.08–2.18)** | **80.75** | **45.57** |
| **hsa-miR-1180** | **0.013** | **1.56 (1.1–2.23)** | **69.57** | **45.93** |
| **hsa-miR-3127** | **0.0055** | **1.65 (1.15–2.35)** | **83.57** | **45.93** |
| **hsa-miR-3677** | **0.0000015** | **2.42 (1.67–3.51)** | **80.75** | **37.32** |
| | | Decreased expression in HCC | | |
| hsa-miR-542 | 0.8858 | 0.97 (0.69–1.38) | 60.89 | 55.69 |
| hsa-miR-214 | 0.8781 | 1.03 (0.73–1.45) | 70.06 | 53.33 |
| hsa-miR-29c | 0.8026 | 0.96 (0.68–1.35) | 81.73 | 55.4 |
| hsa-miR-199b | 0.7972 | 1.05 (0.74–1.48) | 60.89 | 55.4 |
| hsa-miR-33b | 0.5578 | 1.11 (0.78–1.57) | 69.57 | 53.33 |
| hsa-miR-3614 | 0.5502 | 0.9 (0.64–1.27) | 60.89 | 55.69 |
| hsa-miR-10a | 0.4714 | 0.88 (0.62–1.25) | 70.06 | 55.4 |
| hsa-miR-497 | 0.4714 | 0.88 (0.62–1.25) | 60.89 | 55.4 |
| hsa-miR-26b | 0.4599 | 1.14 (0.8–1.62) | 107.11 | 48.99 |
| hsa-miR-424 | 0.3057 | 1.2 (0.85–1.7) | 81.73 | 51.29 |
| hsa-let-7c | 0.166 | 0.78 (0.55–1.11) | 40.41 | 69.57 |
| hsa-miR-195 | 0.1263 | 0.76 (0.54–1.08) | 53.39 | 58.88 |
| hsa-miR-130a | 0.1166 | 0.76 (0.53–1.07) | 51.29 | 69.57 |
| hsa-miR-3607 | 0.1029 | 0.75 (0.53–1.06) | 45.57 | 70.06 |
| **hsa-miR-3653** | **0.0376** | **0.69 (0.49–0.98)** | **41.79** | **80.75** |
| **hsa-miR-99a** | **0.0333** | **0.68 (0.48–0.97)** | **41.79** | **70.06** |
| **hsa-miR-145** | **0.0326** | **0.68 (0.48–0.97)** | **46.78** | **69.57** |
| **hsa-miR-326** | **0.0002** | **1.97 (1.37–2.83)** | **81.73** | **45.57** |
| **hsa-miR-139** | **0.00000057** | **0.41 (0.28–0.59)** | **30.61** | **80.75** |

Abbreviations: miRNA, microRNA; HCC, hepatocellular carcinoma; HR, hazard ratio; OS, overall survival.

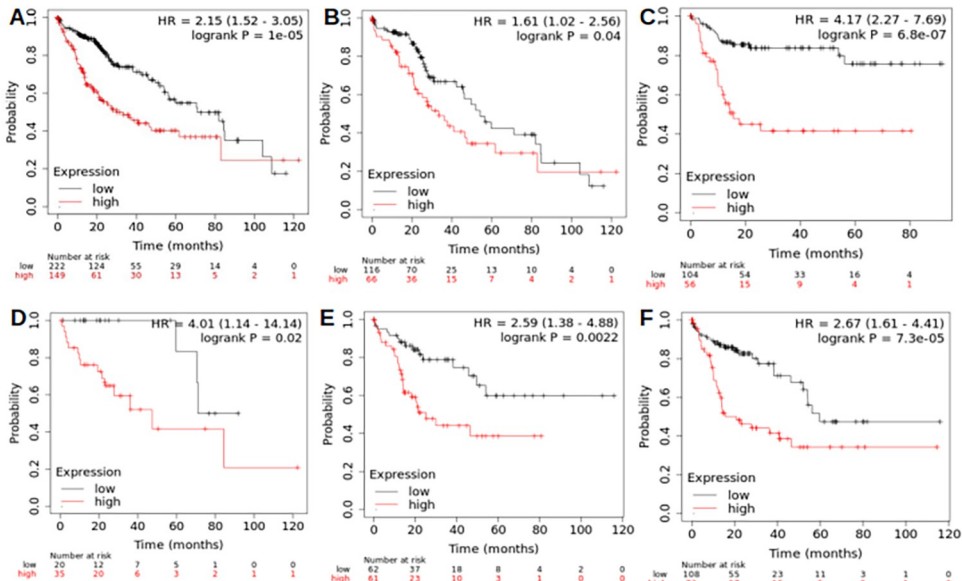

**Fig 1.** Nine-miRNA prognostic signature predicting overall survival in the overall HCC cohort (A) and sub-groups: White-Caucasians (B), Asians (C), well-differentiated (D) and poorly-differentiated HCCs (E), high-TMB HCCs (F).

Taken together, these results suggest that the 5 miRNAs up-regulated in HCC and associated with worse prognosis may be oncogenic, whilst the 4 miRNAs that are down-regulated in HCC and associated with better prognosis may act as tumour suppressors. Based on these criteria, we could not predict a mechanistic role for *Hsa-miR*-326. For this reason, we decided to also study the functional pathways associated with this peculiar transcript.

## 3.2 Functional pathways of miRNA-targeted genes

To understand which molecular pathways are modulated by differential miRNA expression in HCC, we used a dataset of validated miRNA/mRNA target pairs. This enabled us to identify potential functional differences between the two types of miRNAs. A total of 982 unique target protein-coding genes for the 10 miRNAs were retrieved from MiRPathDB. Among these, 430 genes were targeted by up-regulated miRNAs, while 585 genes were targeted by down-regulated ones. We then ran pathway analyses on the two separate lists of protein coding genes.

Pathway analysis on target genes was conducted with DAVID and Webgestalt. Target genes of up-regulated miRNAs were significantly enriched in 7 (DAVID) and 1 (Webgestalt) KEGG pathways. The associated pathways were mainly implicated in neurodegenerative diseases. After filtering for HCC-specific pathways, cell cycle was the only pathway associated with targets of up-regulated miRNAs in both DAVID and Webgestalt tools (**Table 2**).

**Table 2.  HCC-specific pathways relative to up-regulated miRNAs.**

| Term (KEEG) | Genes | % | P-Value | Fold Enrichment | Benjamini | FDR |
|---|---|---|---|---|---|---|
| hsa04110:Cell cycle | 12 | 2.790698 | 0.000356 | 3.716564 | 0.019756 | 0.019392 |

Abbreviations: FDR, false discovery rate.

**Table 3. HCC-specific pathways relative to down-regulated miRNAs.**

| Term (KEGG) | Genes | % | P-Value | Fold Enrichment | Benjamini | FDR |
|---|---|---|---|---|---|---|
| hsa05200:Pathways in cancer | 58 | 9.7643 | 3.41E-12 | 2.71604428 | 2.11E-10 | 1.44E-10 |
| hsa05225:Hepatocellular carcinoma | 30 | 5.0505 | 1.73E-11 | 4.44033101 | 7.66E-10 | 5.22E-10 |
| hsa05161:Hepatitis B | 25 | 4.2088 | 2.39E-08 | 3.83732310 | 3.91E-07 | 2.66E-07 |
| hsa04010:MAPK signaling pathway | 34 | 5.7239 | 6.48E-08 | 2.87564294 | 8.74E-07 | 5.95E-07 |
| hsa04151:PI3K-Akt signaling pathway | 36 | 6.0606 | 5.86E-07 | 2.52873088 | 6.27E-06 | 4.27E-06 |
| hsa05206:MicroRNAs in cancer | 32 | 5.3872 | 2.11E-06 | 2.56679780 | 2.05E-05 | 1.39E-05 |
| hsa04110:Cell cycle | 16 | 2.6936 | 0.000146 | 3.15756872 | 8.54E-04 | 5.81E-04 |
| hsa05160:Hepatitis C | 18 | 3.0303 | 0.000174 | 2.85086220 | 9.65E-04 | 6.57E-04 |
| hsa04350:TGF-beta signaling pathway | 12 | 2.0202 | 0.001279 | 3.17436430 | 5.43E-03 | 3.70E-03 |
| hsa04115:p53 signaling pathway | 9 | 1.5152 | 0.008519 | 3.06565319 | 3.01E-02 | 2.05E-02 |

Abbreviations: FDR, false discovery rate.

Targets of down-regulated miRNAs were significantly enriched in 93 (DAVID) and 94 (Webgestalt) KEGG pathways relating to numerous cancers, cancer related processes and associated signalling pathways. Ten HCC-specific pathways were significantly enriched ($p<0.05$), including HCC, Hepatitis B, MAPK signalling pathway, PI3K-Akt signalling pathway, Cell cycle, Hepatitis C, TGF-beta signalling and p53 signalling pathway (**Table 3**).

A graphical representation of the results of the above-mentioned enrichment analyses is presented in **Fig 2**.

A Comprehensive overview of KEGG HCC-specific pathways can be found in **S1 Fig**.

The process of intersecting the miRNA target genes with the HCC and pan cancer associated genes revealed that the list of 982 miRNA target genes contained 10 HCC driver genes and 11 pan-cancer driver genes (**Table 4**).

To further validate the clinical significance of our findings, we studied the predictive value of HCC-specific miRNA-targeted protein coding genes (Table 4) in a small but well characterised cohort of HCC patients treated with sorafenib. Notably, higher expression of 3 miRNA-targeted genes (*RB1*, *NOTCH2*, *PIK3CA*) was associated with better survival (Fig 3A–3C). In

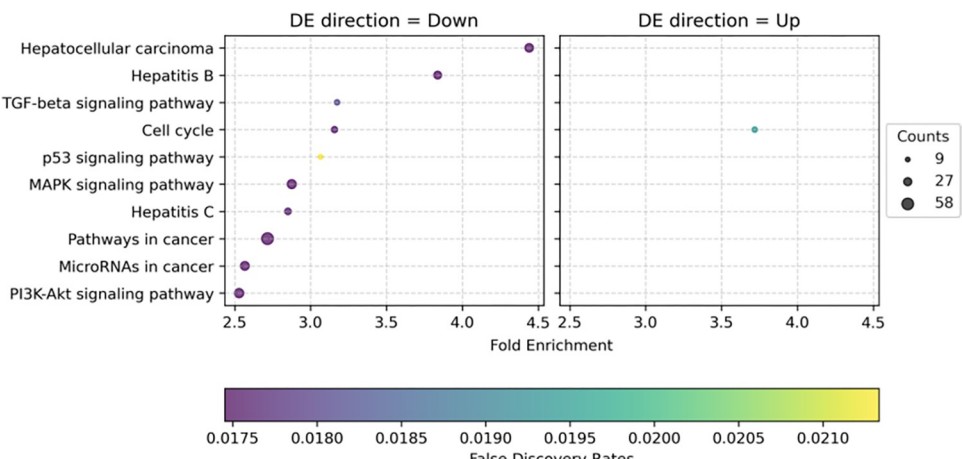

**Fig 2. Enrichment analysis for miRNA-targeted genes in KEGG pathways.** The shared x-axis represents fold enrichment and False Discovery Rates (FDR) for both plots. All listed KEGG pathways represent the enrichment of down-regulated miRNAs (left plot) and up-regulated miRNAs (right plot) targeting genes in different pathways.

**Table 4. HCC- and pan-cancer driver genes among miRNA target genes.**

| Symbol | Ensembl Gene ID | Chr | Targeting miRNAs | Position (Mbp) | Description |
|---|---|---|---|---|---|
| THRAP3 | ENSG00000054118 | 1 | hsa-miR-877-5p, hsa-miR-3127-5p, hsa-miR-326, hsa-miR-145-5p, hsa-miR-99a-5p | 36.2244 | thyroid hormone receptor associated protein 3 |
| NRAS (*) | ENSG00000213281 | 1 | hsa-mir-501-3p, hsa-miR-877-5p, hsa-miR-139-5p, hsa-miR-145-5p, hsa-miR-326 | 114.7045 | NRAS proto-oncogene, GTPase |
| NOTCH2 | ENSG00000134250 | 1 | hsa-miR-139-5p, hsa-miR-326, hsa-miR-3127-5p, hsa-miR-3677-3p, hsa-miR-145-5p | 119.9116 | notch receptor 2 |
| TBL1XR1 | ENSG00000177565 | 3 | hsa-miR-139-5p, hsa-mir-501-3p, hsa-miR-326, hsa-miR-145-5p, hsa-miR-3677-3p, hsa-miR-877-5p, hsa-miR-3127-5p | 177.0193 | TBL1X receptor 1 |
| PIK3CA (*) | ENSG00000121879 | 3 | hsa-mir-501-3p, hsa-miR-326, hsa-miR-139-5p, hsa-miR-145-5p, hsa-miR-877-5p | 179.1481 | phosphatidylinositol-4,5-bisphosphate 3-kinase catalytic subunit alpha |
| MSH3 | ENSG00000113318 | 5 | hsa-mir-501-3p, hsa-miR-139-5p, hsa-miR-877-5p, hsa-miR-326, hsa-miR-145-5p | 80.6547 | mutS homolog 3 |
| H1-4 (*) | ENSG00000168298 | 6 | hsa-miR-1180-3p | 26.1563 | H1.4 linker histone, cluster member |
| CDKN1A (*) | ENSG00000124762 | 6 | hsa-miR-3127-5p, hsa-miR-877-5p, hsa-miR-145-5p | 36.6765 | cyclin dependent kinase inhibitor 1A |
| EEF1A1 (*) | ENSG00000156508 | 6 | hsa-miR-877-5p | 73.4893 | eukaryotic translation elongation factor 1 alpha 1 |
| ESR1 | ENSG00000091831 | 6 | hsa-mir-501-3p, hsa-miR-3127-5p, hsa-miR-99a-5p, hsa-miR-877-5p, hsa-miR-3677-3p, hsa-miR-145-5p, hsa-miR-326, hsa-miR-139-5p | 151.6567 | estrogen receptor 1 |
| MYC | ENSG00000136997 | 8 | hsa-miR-145-5p | 127.7354 | MYC proto-oncogene, bHLH transcription factor |
| KMT2A | ENSG00000118058 | 11 | hsa-miR-139-5p, hsa-miR-877-5p, hsa-miR-1180-3p, hsa-miR-3127-5p, hsa-miR-326 | 118.4365 | lysine methyltransferase 2A |
| KRAS (*) | ENSG00000133703 | 12 | hsa-miR-877-5p, hsa-mir-501-3p, hsa-miR-3127-5p, hsa-miR-1180-3p, hsa-miR-145-5p, hsa-miR-326 | 25.2052 | KRAS proto-oncogene, GTPase |
| RB1 (*) | ENSG00000139687 | 13 | hsa-miR-3127-5p, hsa-miR-99a-5p, hsa-miR-326 | 48.3037 | RB transcriptional corepressor 1 |
| CHD8 | ENSG00000100888 | 14 | hsa-miR-877-5p | 21.3852 | chromodomain helicase DNA binding protein 8 |
| BRD7 (*) | ENSG00000166164 | 16 | hsa-miR-3127-5p, hsa-miR-145-5p, hsa-mir-501-3p, hsa-miR-326, hsa-miR-3677-3p, hsa-miR-99a-5p | 50.3135 | bromodomain containing 7 |
| BCL2 | ENSG00000171791 | 18 | hsa-miR-145-5p, hsa-miR-3127-5p, hsa-miR-99a-5p, hsa-miR-139-5p, hsa-mir-501-3p | 63.1233 | BCL2 apoptosis regulator |
| SMARCA4 (*) | ENSG00000127616 | 19 | hsa-miR-139-5p | 10.9609 | SWI/SNF related, matrix associated, actin dependent regulator of chromatin, subfamily a, member 4 |
| RPS6KA3 (*) | ENSG00000177189 | X | hsa-miR-139-5p, hsa-miR-145-5p, hsa-miR-326, hsa-miR-877-5p | 20.1499 | ribosomal protein S6 kinase A3 |
| USP9X | ENSG00000124486 | X | hsa-miR-1180-3p, hsa-miR-877-5p, hsa-mir-501-3p, hsa-miR-145-5p, hsa-miR-139-5p | 41.0854 | ubiquitin specific peptidase 9 X-linked |
| AR | ENSG00000169083 | X | hsa-miR-139-5p, hsa-miR-99a-5p, hsa-miR-877-5p, hsa-miR-145-5p, hsa-miR-326, hsa-miR-3127-5p | 67.544 | androgen receptor |

(*) refer to HCC-specific genes.

each case, higher expression of the protein coding genes was a positive predictive factor (longer survival in sorafenib-treated patients). The three protein coding genes are targeted by 3 to 5 of our investigated miRNAs (Table 4, fourth column). Notably, the only miRNA that targets all these three protein-coding genes is *Hsa-miR*-326, which is down regulated in HCC, but whose higher expression is associated with worse prognosis. Moreover, the combined three-gene signature achieved an even greater predictive significance (Fig 3D).

Thus, there is experimental evidence that the investigated miRNAs may also directly interact with 21 putative HCC and pan-cancer driver genes.

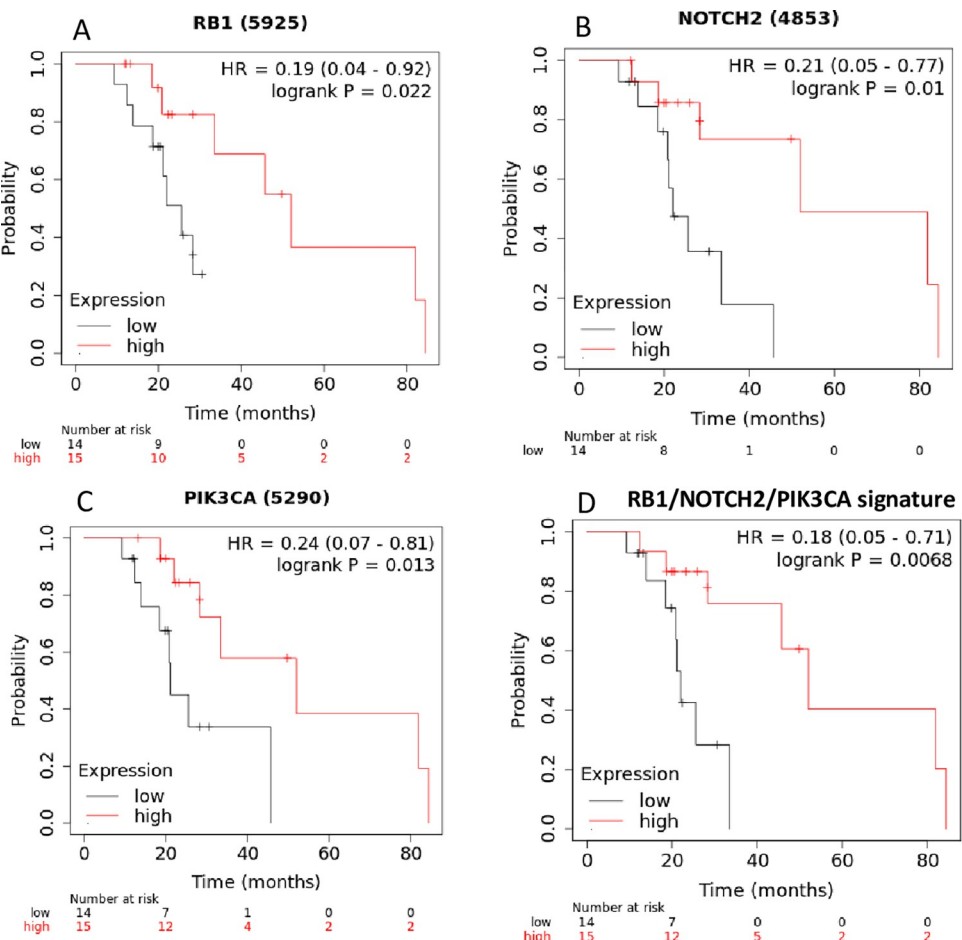

**Fig 3. HCC-specific miRNA-targeted genes significantly associated with overall survival in the sorafenib-treated cohort from Kaplan Meier plotter.** (A) *RB1;* (B) *NOTCH2*; (C) *PIK3CA*. (D) Kaplan Meier plot for the 3-genes signature in the same population.

## 4. Discussion

In this study, we have identified a 9-miRNA prognostic signature for HCC. The proposed signature is consistently significant across different clinically relevant HCC sub-groups (e.g. ethnicity, nuclear differentiation grade) and seems to have a prominent prognostic role among Asian patients, possibly due to more frequent HBV-aetiology in this ethnic group [20]. In keeping with this hypothesis, HBV-aetiology is the second most significant HCC-pathway affected by our miRNA signature (Fig 2). If confirmed in clinical datasets, this signature could enable patient stratification and treatment optimisation, especially for low- and intermediate-risk diseases. At these stages, several therapeutic options are available, but currently these treatments cannot be tailored based on individual molecular profiles. Interestingly, some miRNAs are well detectable in biological fluids, such as plasma [21]. If the 9-miRNA signature proves to be detectable in plasma samples in the HCC population, it can be used as a minimally invasive biomarker for patient stratification and dynamic monitoring.

Several experimental studies have dissected the pathogenic role of specific miRNAs in HCC. Our results are in line with mechanistic evidence. Of the five up-regulated/negatively prognostic miRNAs, four have been shown to act as oncogenes in HCC cells: *mir-501* by

promoting epithelial to mesenchymal transition [22]; *mir-3127* and *mir-1180* by promoting proliferation and tumorigenesis [23, 24]; *mir-3677* by promoting proliferation and drug resistance [25]. While the clinical significance of some of these transcripts has been described, to the best of our knowledge this is the first study showing that mir-3127 has a prognostic impact on HCC. Notably, our pathway analysis has identified cell cycle regulation as the main molecular pathway controlled by these miRNAs, further confirming the link between bioinformatic and experimental findings.

Our results on *mir-877* are apparently in contrast with previous reports, showing that this transcript inhibits HCC proliferation [26]. We found that this miRNA is up-regulated in HCC *versus* normal tissue, and that higher expression of this miRNA predicts poor prognosis. Both results imply an oncogenic function. Due to these coherent results, we decided to maintain *mir-877* in our prognostic signature. Notably, another study showed that a genetic variant of this miRNA predicts HCC risk [27]. It is not known if this genetic variant has a prognostic role in advanced disease settings. It is therefore conceivable that this miRNA could play different roles at different stages of disease progression (e.g. onco-suppressive for early stage, oncogenic for advanced stage), as demonstrated for other cancer-related genes [28]. In light of the partially conflicting evidence, further studies are warranted to dissect the role of this miRNA at different stages of HCC progression, and to test its prognostic role in other clinical datasets.

In parallel with the results on putative oncogenic miRNAs, our results on putative tumour suppressors agree with pre-clinical results. *Mir-3653*, *mir-326* and *mir-139* have been shown to inhibit metastasis and progression of HCC cells [29–31]. Similarly, *mir-145* has been implicated in G2/M phase arrest by inhibiting cyclin B1 [32]. Finally, a recent study has shown that *mir-99a* is a direct target of the epigenetic silencer EZH2 [33]. EZH2 promotes cell proliferation, metastasis and drug resistance in several cancers by silencing the expression of tumour suppressors [34]. We have shown that higher EZH2 activity (measured via a liquid biopsy approach) predicts poorer prognosis in HCC patients treated with sorafenib [35]. Hence our finding on *mir-99a* paves the way for the combined use of miRNA and epigenetic biomarkers for patient stratification and drug efficacy prediction in HCC.

Other recent manuscripts have investigated miRNA signatures in HCC. For example, Sathipati et al identify a 23-miRNA signature associated with stage [10]. Notably, 7 of these miRNAs had a prognostic role, based on the analysis of a dataset of 166 HCC samples. We acknowledge that some aspects of the methodology overlap with our study. However, there are also substantial differences between the two studies. Sathipati et al have used machine learning for miRNA selection, focussing on transcript expression at different cancer stages, whilst we have manually filtered the transcripts based on differential expression in HCC vs normal tissue and we have subsequently refined our signature based on prognostic value. Importantly, we have validated our prognostic signature in a larger dataset containing data from 372 HCC samples. Our 9-miRNA signature does not contain any of the 7 miRNAs identified by Sathipati et al. This divergence is attributable to the substantially different methodologies of the two studies. Interestingly 3 out of 5 HCC-specific upregulated miRNAs have been measured in biological fluids from cancer patients [36–38]. This result suggests the potential clinical relevance of our miRNA signature to guide clinical diagnosis and prognosis.

We also investigated the predictive role of miRNA-modulated pathways in a curated cohort of HCC patients exposed to the tyrosine kinase inhibitor sorafenib. Sorafenib is employed for the treatment of advanced HCC as second line treatment after progression on first-line immunotherapy-based strategies, or as front-line choice for patients with absolute contraindications to combination strategies. Our results showed that higher expression of three target genes was associated with better response to the treatment. Notably, the only miRNA that targets all these three protein-coding genes is *mir-326*, which is down regulated in HCC, but whose

*higher* expression is associated with worse prognosis. Because of this intriguing inconsistence, we decided to perform pathway analyses on this transcript. Based on our results, it is conceivable that this miRNA could play different roles at different stages of HCC pathogenesis. Whilst inhibiting early tumorigenic events, *mir-326* could suppress the expression of therapy-sensitivity genes (*NOTCH*, *RB1*, *PIK3CA*) thereby conferring sorafenib resistance to HCC cells. These findings highlight the potential predictive role of the miRNA signature and of the associated target mRNAs.

In conclusion, we have identified for the first time a miRNA signature that is differentially expressed in HCC and significantly predicts prognosis. This signature includes transcripts whose function has been independently validated in pre-clinical studies. Future studies on clinical samples (primary tumours and plasma) are warranted to confirm the clinical usefulness of this potential new biomarker panel.

## Supporting information

**S1 Table. Number of experimental reads of the predominant mature form of each miRNA.**
(DOCX)

**S2 Table. Fifty-nine differentially expressed miRNAs (2-fold) in HCC vs normal livers according to OMCD.**
(XLSX)

**S1 Fig. KEGG HCC pathways.** Enriched genes from target up- or down-regulated miRNAs are red-highlighted.
(TIF)

**S2 Fig. Kaplan Meier plots showing prognostic significance for OS of each miRNA of the signature.**
(TIF)

## Author Contributions

**Conceptualization:** Kayleigh Smith, Dan Beach, Francesca Salani, Francesco Crea.

**Data curation:** Kayleigh Smith, Dan Beach, Roger Silva, Gyorffy Balazs, Francesca Salani, Francesco Crea.

**Formal analysis:** Dan Beach, Roger Silva.

**Investigation:** Francesca Salani.

**Methodology:** Roger Silva, Gyorffy Balazs, Francesca Salani.

**Resources:** Gyorffy Balazs.

**Software:** Roger Silva.

**Supervision:** Francesca Salani, Francesco Crea.

**Validation:** Francesca Salani, Francesco Crea.

**Visualization:** Francesca Salani.

**Writing – original draft:** Kayleigh Smith, Dan Beach, Roger Silva, Francesca Salani, Francesco Crea.

**Writing – review & editing:** Kayleigh Smith, Dan Beach, Roger Silva, Gyorffy Balazs, Francesca Salani, Francesco Crea.

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
