## [Decision Letter · Decision Letter 0]

3 Jan 2024

PONE-D-23-40355Comprehensive analysis of differentially expressed miRNAs in Hepatocellular carcinoma: prognostic significance and pathway insightsPLOS ONE

Dear Dr. Salani,

Thank you for submitting your manuscript to PLOS ONE. After careful consideration, we feel that it has merit but does not fully meet PLOS ONE’s publication criteria as it currently stands. Therefore, we invite you to submit a revised version of the manuscript that addresses the points raised during the review process. Authors need to address following concerns:1. There is a similar study on miRNA profile in HCC. please address how this study is different from published  one. In particular,  need to address those novel miRNAs identified in this study.2.  It is not clear how important those miRNAs identified  and correlated with clinical signatures including overall or progression free survival. Is there any evidence to show they are potential biomarkers for diagnosis or prognosis.

We look forward to receiving your revised manuscript.

Kind regards,

Junming Yue

Academic Editor

PLOS ONE

Journal Requirements:

2. Please note that your Data Availability Statement is currently missing [the repository name and/or the DOI/accession number of each dataset OR a direct link to access each database]. If your manuscript is accepted for publication, you will be asked to provide these details on a very short timeline. We therefore suggest that you provide this information now, though we will not hold up the peer review process if you are unable.

Reviewers' comments:

Reviewer's Responses to Questions

**Comments to the Author**

1. Is the manuscript technically sound, and do the data support the conclusions?

Reviewer #1: Yes

Reviewer #2: Partly

2. Has the statistical analysis been performed appropriately and rigorously? 

Reviewer #1: Yes

Reviewer #2: I Don't Know

3. Have the authors made all data underlying the findings in their manuscript fully available?

Reviewer #1: Yes

Reviewer #2: Yes

4. Is the manuscript presented in an intelligible fashion and written in standard English?

Reviewer #1: Yes

Reviewer #2: Yes

5. Review Comments to the Author

Reviewer #1: In this study, the authors studied the miRNAs in hepatocellular carcinoma to identify a set of miRNA that can potentially be used for HCC prognosis prediction. While some of these miRNAs have already been reported in literature to be associated with HCC, the authors were generally transparent about this and provided clear comparisons between their results versus the reported ones. Overall I found this manuscript of potential interest to the readers of this journal and could provide some value to researchers in this field. However, the following comments need to be addressed:

1. In the title there is no need to capitalize the first letter of "Hepatocellular"

2. What does "co-last authors" mean? I thought it means co-corresponding, but only one of them is labeled as the corresponding author, so I am a bit confused.

3. line 31, the authors claimed they identified "novel miRNAs" with is misleading since most of their reported miRNAs have already been shown in literature to be associated with HCC. Please revise this sentence to be more objective.

4. line 53 "bioinformatic" is a single world

5. line 108 please specify how many genes have been removed in this step, and how many are left to proceed with next steps.

6. Section 2.4.5 needs more details. Please specify how many pathways were identified to be enriched by both methods, and how many are not. Did the authors proceed with only the ones enriched in both methods?

7. line 232-240, it is interesting that the authors found the role of miR-877 to be contradicting with some literature. The authors provided a plausible hypothesis that miR-877 could be playing opposite roles in different stages of HCC. However, following this hypothesis, should we still consider miR-877 as a prognostic biomarker for HCC? The authors should comment on this in their discussion based on their own study and analysis.

8. line 250-251, I think it is too assertive to claim that these 9-miRNA signature "robustly predicts prognosis" since this study solely relies on pubic database with no additional experimental validation.

Reviewer #2: The manuscript by K Smith analyzed miRNA data of HCC patients from public database (TCGA) and identified a 9-miRNA signature for HCC patient’s prognosis. The authors also did some pathway analysis and compared their findings with the published HCC and pan cancer associated genes. Here are some concerns:

1. Novelty. This manuscript is very similar to an article published in 2020 (DOI: https://doi.org/10.1038/s41598-020-71324-z), which identified a 23-miRNA signature for HCC stage and prognosis using machine learning. The authors did not discuss this publication in this manuscript.

2. The authors compared the miRNA target genes with published cancer associated gene set, and then concluded that there are ‘direct miRNA interactions with cancer driver genes’, which is not sound enough for the conclusion.

3. Figure 1B and 1C show big differences between White-Caucasians and Asians patients. It would be more interesting and meaningful if the authors discussed the possible reasons.

4. It is worth to show the survival rate of patients related with each miRNA in the 9 miRNAs.

6. PLOS authors have the option to publish the peer review history of their article (what does this mean?). If published, this will include your full peer review and any attached files.

Reviewer #1: No

Reviewer #2: No

---

## [Author Response · Author response to Decision Letter 0]

22 Feb 2024

All the information are given in the rebuttal letter uploaded. Please find the same information cut and pasted here as well.

We thank the Editor for having considered our work for publication in PlosOne journal, and we would like to thank the Reviewers for their valuable suggestions.

Herein we provide a specific rebuttal for each point raised by the two Reviewers and the general comments of the Editor.

Editor’s comments

1. There is a similar study on miRNA profile in HCC. please address how this study is different from published one. In particular, need to address those novel miRNAs identified in this study.

We have now extensively discussed the study and the substantial novelty of our findings. Please see response “I” (to Reviewer 2) and associated amendments to the manuscript. 

2. It is not clear how important those miRNAs identified and correlated with clinical signatures including overall or progression free survival. Is there any evidence to show they are potential biomarkers for diagnosis or prognosis.

Our study has identified miRNAs that are up-regulated in HCC and prognostically relevant. These two aspects are crucial for diagnosis and prognosis. To highlight the clinical relevance of this signature, we have now discussed the feasibility of detecting these transcripts in biological fluids (see lines 285-288 and references 35-37).

To further increase the clinical relevance of our findings, we have investigated whether any of the predicted mRNA target genes is associated with response to sorafenib in HCC patients (Figure 3). As discussed, sorafenib is currently used in the treatment of this malignancy. 

Unfortunately, we could not study the predictive role of miRNAs in sorafenib-treated patients as the dataset only contained protein-coding genes. Nonetheless, we think that these two amendments will increase the clinical relevance of our study. 

Reviewer 1:

A) In the title there is no need to capitalize the first letter of "Hepatocellular"

Edited as suggested.

B) What does "co-last authors" mean? I thought it means co-corresponding, but only one of them is labelled as the corresponding author, so I am a bit confused.

During the process of submission, we had asked the Editor if it would have been possible to identify two co-last Authors (similarly to the two co-primary ones), since Crea and Salani had contributed equally to study conception, student supervision, data interpretation and critical revision. This possibility is granted by some international journals, but we had not received a clear reply from the Editor in this regard. If not allowed by PlosOne, we will edit according to the Editor’s recommendation.

C) line 31, the authors claimed they identified "novel miRNAs" which is misleading since most of their reported miRNAs have already been shown in literature to be associated with HCC. Please revise this sentence to be more objective.

We acknowledge that each of the 10 miRNAs has been already identified as HCC-related, mainly in pre-clinical studies. We discuss previously published evidence in lines 248-275, whilst noting that the prognostic role of some transcripts is novel (see for example new added sentence in line 252-254). In addition, we would like to point out that these transcripts have never been comprehensively included in a single prognostic HCC signature (https://doi.org/10.1093/carcin/bgad062;
https://doi.org/10.1371/journal.pone.0128628;https://doi.org/10.1002/hep.22160). 

Considering the Reviewer’s suggestion, we rephrased “novel miRNAs” with “a novel miRNA signature”.

D) line 53 "bioinformatic" is a single world

Edited as suggested.

E) line 108 please specify how many genes have been removed in this step, and how many are left to proceed with next steps.

There was a 100% match between target genes in the mirPathDB and mirTargetLink database, so it was not necessary to remove any genes at this stage. We have now specified this in Methods (current line 109).

F). Section 2.4.5 needs more details. Please specify how many pathways were identified to be enriched by both methods, and how many are not. Did the authors proceed with only the ones enriched in both methods?

We thank the Reviewer for this important question, which allowed us to clarify our methodology. In the Results section (lines 138, 187-191, 194) we now specify how many pathways were present in both datasets and which ones were overlapping. 

G) Line 232-240, it is interesting that the authors found the role of miR-877 to be contradicting with some literature. The authors provided a plausible hypothesis that miR-877 could be playing opposite roles in different stages of HCC. However, following this hypothesis, should we still consider miR-877 as a prognostic biomarker for HCC? The authors should comment on this in their discussion based on their own study and analysis.

We thank the Reviewer for the point raised. As requested, we have explained why we have maintained this miRNA in our prognostic signature: we found internally coherent results in our datasets (up-regulation in HCC vs normal; negative prognostic value. See line 257-266). We acknowledge that further studies are needed to elucidate miR-877 function in HCC and to “test” (not to “confirm” as previously stated) its prognostic role in independent datasets.

H) Line 250-251, I think it is too assertive to claim that these 9-miRNA signature "robustly predicts prognosis" since this study solely relies on public database with no additional experimental validation.

We thank the Reviewer for the suggestion. We amended the sentence to “significantly predict prognosis”.

Reviewer 2:

The manuscript by K Smith analyzed miRNA data of HCC patients from public database (TCGA) and identified a 9-miRNA signature for HCC patient’s prognosis. The authors also did some pathway analysis and compared their findings with the published HCC and pan cancer associated genes. Here are some concerns:

I) Novelty. This manuscript is very similar to an article published in 2020 (DOI: https://doi.org/10.1038/), which identified a 23-miRNA signature for HCC stage and prognosis using machine learning. The authors did not discuss this publication in this manuscript.

We thank the Reviewer for highlighting this interesting study, which is now cited in the Introduction (line 53) and extensively discussed in lines 276-288.

To further increase the novelty of our findings, we have explored whether the protein-coding genes targeted by our prognostic miRNAs were associated with sorafenib response (Figure 3). This correlation had been never investigated in similar studies. This analysis was enabled by the availability of a small but well curated dataset that contained only mRNA genes. Our results are interesting and provide potential explanations on the prognostic role of mir-326 (see lines 295-300).

J) The authors compared the miRNA target genes with published cancer associated gene set, and then concluded that there are ‘direct miRNA interactions with cancer driver genes’, which is not sound enough for the conclusion.

We apologies for the over-statement, and we agree with the Reviewer that we only performed a bioinformatic prediction. To reflect this, we have changed the title of section 2.4.6 which is now:” Identifying potential miRNA interactions with cancer driver genes”. 

K) Figure 1B and 1C show big differences between White-Caucasians and Asians patients. It would be more interesting and meaningful if the authors discussed the possible reasons.

The substantial overall survival difference between Asian and Caucasian patients is probably due to divergent HCC aetiology as highlighted by many epidemiological studies: HBV-related (Asians) vs HCV or non-viral aetiology (Caucasians). See this study for reference https://doi.org/10.1111/liv.15251, 10.4254/wjh.v7.i12.1708. We now discuss this aspect in lines 237-241.

L) It is worth to show the survival rate of patients related with each miRNA in the 9 miRNAs.

Kaplan Meier plots of overall survival associated with the 9 different miRNAs have been added to the supplementary material, as Figure S2 (lines 173-174).

---

## [Decision Letter · Decision Letter 1]

6 Mar 2024

PONE-D-23-40355R1Comprehensive analysis of differentially expressed miRNAs in hepatocellular carcinoma: prognostic, predictive significance and pathway insightsPLOS ONE

Dear Dr. Salani,

Thank you for submitting your manuscript to PLOS ONE. After careful consideration, we feel that it has merit but does not fully meet PLOS ONE’s publication criteria as it currently stands. Therefore, we invite you to submit a revised version of the manuscript that addresses the points raised during the review process.

We look forward to receiving your revised manuscript.

Kind regards,

Junming Yue

Academic Editor

PLOS ONE

Journal Requirements:

**Additional Editor Comments:**==============================Please reconcile miR-326 with 10 miRNA signature throughout the manuscript and have to be consistent including in discussion.

Reviewers' comments:

Reviewer's Responses to Questions

**Comments to the Author**

1. If the authors have adequately addressed your comments raised in a previous round of review and you feel that this manuscript is now acceptable for publication, you may indicate that here to bypass the “Comments to the Author” section, enter your conflict of interest statement in the “Confidential to Editor” section, and submit your "Accept" recommendation.

Reviewer #2: All comments have been addressed

2. Is the manuscript technically sound, and do the data support the conclusions?

Reviewer #2: Partly

3. Has the statistical analysis been performed appropriately and rigorously? 

Reviewer #2: Yes

4. Have the authors made all data underlying the findings in their manuscript fully available?

Reviewer #2: Yes

5. Is the manuscript presented in an intelligible fashion and written in standard English?

Reviewer #2: Yes

6. Review Comments to the Author

Reviewer #2: One concern now is that the authors stated they found a 9--miRNA signature for HCC based on their hypothesis, miRNA-326 was removed because of its unexpected correlation with prognosis. However, the following study of pathways and miRNA-target genes all included miRNA-326 (line 184), which makes it very confusing. It also says 10-miRNA signature in the discussion (line 285), which needs to be clarified. The pathways and target genes of niRNA-326 could be studied and discussed separately since it has opposite correlation with prognosis than expected.

7. PLOS authors have the option to publish the peer review history of their article (what does this mean?). If published, this will include your full peer review and any attached files.

Reviewer #2: No

---

## [Author Response · Author response to Decision Letter 1]

18 Mar 2024

We would like to thank the Editor and Reviewer2 for pointing out the inconsistency in the miRNA-signature definition. We have now carefully reviewed the manuscript, making sure that the phrase “prognostic signature” is applied only to the group of 9 (not 10) miRNAs. We hope this will reduce confusion.

In this study, we have also tried to investigate potential molecular pathways associated with notable miRNAs. For this mechanistical analysis, we decided to keep the “odd” miRNA that showed opposite correlation between prognosis and expression patterns in normal vs neoplastic liver tissue (mir-326). In the new version of the manuscript, we have made this distinction clear (lines 179-188). As suggested by the reviewer, this miRNA has been discussed separately (lines 300-305). 

In the process of this further manuscript’s revision, we also edited:

- A missing reference in line 242 (added between bracket)

- A typo in line 295

- . instead of , in line 32-33

- Patients’ instead of patients, in line 34

- Pathway instead of pathways, in line 45

- “The Cancer Genome Atlas (TCGA)” was misspelled through out the text and not it has been reconciled.

- Ran instead of run, in line 189

- And removed from line 290

---

## [Editor Report · Decision Letter 2]

20 Mar 2024

Comprehensive analysis of differentially expressed miRNAs in hepatocellular carcinoma: prognostic, predictive significance and pathway insights

PONE-D-23-40355R2

Dear Dr. Salani,

We’re pleased to inform you that your manuscript has been judged scientifically suitable for publication and will be formally accepted for publication once it meets all outstanding technical requirements.

An invoice for payment will follow shortly after the formal acceptance. To ensure an efficient process, please log into Editorial Manager at Editorial Manager® , click the 'Update My Information' link at the top of the page, and double check that your user information is up-to-date. If you have any billing related questions, please contact our Author Billing department directly at authorbilling@plos.org.

Kind regards,

Junming Yue

Academic Editor

PLOS ONE

---

## [Editor Report · Acceptance letter]

27 Mar 2024

PONE-D-23-40355R2 

PLOS ONE

Dear Dr. Salani, 

I'm pleased to inform you that your manuscript has been deemed suitable for publication in PLOS ONE. Congratulations! Your manuscript is now being handed over to our production team.

Kind regards, 

on behalf of

Dr. Junming Yue 

Academic Editor

PLOS ONE